# Real-Life Considerations on Antifungal Treatment Combinations for the Management of Invasive Mold Infections after Allogeneic Cell Transplantation

**DOI:** 10.3390/jof7100811

**Published:** 2021-09-28

**Authors:** Emmanouil Glampedakis, Romain Roth, Stavroula Masouridi-Levrat, Yves Chalandon, Anne-Claire Mamez, Federica Giannotti, Christian Van Delden, Dionysios Neofytos

**Affiliations:** 1Infectious Diseases Service and Hospital Preventive Medicine Service, Department of Medicine, Lausanne University Hospital, Chemin Mont Paisible 18, CH-1011 Lausanne, Switzerland; 2Division of Infectious Diseases, University Hospital of Geneva, Rue Gabrielle-Perret-Gentil 4, CH-1211 Geneva, Switzerland; Romain.Roth@hcuge.ch (R.R.); christian.vandelden@hcuge.ch (C.V.D.); 3Bone Marrow Transplant Unit, Division of Hematology, University Hospital of Geneva and Faculty of Medicine, University of Geneva, CH-1211 Geneva, Switzerland; Stavroula.Masouridi-Levrat@hcuge.ch (S.M.-L.); yves.chalandon@hcuge.ch (Y.C.); Anne-Claire.Mamez@hcuge.ch (A.-C.M.); Federica.Giannotti@hcuge.ch (F.G.)

**Keywords:** invasive mold infections, invasive aspergillosis, mucormycosis, bone marrow transplantation, allogeneic hematopoietic cell transplantation, antifungal combinations, antifungals, antifungal therapy

## Abstract

Background: Antifungal combination treatment is frequently administered for invasive mold infections (IMIs) after allogeneic hematopoietic cell transplantation (HCT). Here, we describe the indications, timing, and outcomes of combination antifungal therapy in post-HCT IMI. Methods: A single-center, 10-year, retrospective cohort study including all adult HCT recipients with proven/probable IMI between 1 January 2010 and 1 January 2020 was conducted. Results: During the study period, 515 patients underwent HCT, of whom 47 (9.1%) presented 48 IMI episodes (46 patients with one IMI episode and 1 patient with two separate IMI episodes): 33 invasive aspergillosis (IA) and 15 non-IA IMIs. Almost half (51%) of the patients received at least one course of an antifungal combination (median: 2/patient): 23 (49%), 20 (42%), and 4/47 (9%) patients received pure monotherapy, mixed monotherapy/combination, and pure combination treatment, respectively. Combination treatment was started at a median of 8 (IQR: 2, 19) days post-IMI diagnosis. Antifungal management was complex, with 163 treatment courses prescribed overall, 48/163 (29.4%) concerning antifungals in combination. The clinical reasons motivating the selection of initial combination antifungal therapy included severe IMI (18, 38%), lack of antifungal susceptibility data (14, 30%), lack of pathogen identification (5, 11%), and combination treatment until reaching a therapeutic azole serum level (6, 13%). The most common combination treatments were azole/liposomal amphotericin-B (28%) and liposomal amphotericin-B/echinocandin (21%). Combination treatment was administered cumulatively for a median duration of 28 days (IQR: 7, 47): 14 (IQR: 6, 50) days for IA and 28 (IQR: 21, 34) days for non-IA IMI (*p* = 0.18). Overall, 12-week mortality was 30%. Mortality was significantly higher among patients receiving ≥50% of treatment as combination (logrank = 0.04), especially those with non-IA IMI (logrank = 0.03). Conclusions: Combination antifungal treatment is frequently administered in allogeneic HCT recipients with IMI to improve clinical efficacy, albeit in an inconsistent and variable manner, suggesting a lack of relevant data and guidance, and an urgent need for new studies to improve therapeutic options.

## 1. Introduction

Invasive mold infections (IMI) constitute a serious threat for the long-term outcomes of allogeneic hematopoietic cell transplant (HCT) recipients, especially those under continuous immunosuppression due to graft-versus-host disease (GvHD) [1,2,3,4,5]. Invasive aspergillosis (IA) is the most frequent among these infections, followed by non-*Aspergillus* species (non-IA) IMI, with mucormycosis being the predominant infection among the latter in terms of frequency and severity [3,4,5,6,7]. Due to the dismal prognosis of IMI, antifungal combination therapies have been considered, in order to potentially increase efficacy and improve clinical outcomes [8,9,10,11,12,13,14]. However, and despite in vitro and retrospective observational clinical data suggesting a potential benefit, the only prospective, placebo-controlled randomized clinical trial conducted in hematologic patients (including HCT recipients) failed to clearly show a definitive benefit of an azole/echinocandin combination compared to azole monotherapy for the treatment of IA [8,9,14]. In the case of mucormycosis, patients frequently receive multiple antifungals in combination, based on conflicting observational retrospective data, while robust evidence of superiority over polyene monotherapy is lacking both in the first line and salvage context [10,12,13,15,16]. Finally, data on combinations of antifungal drugs for the treatment of other rarer non-IA, non-*Mucorales* IMI are also sparse, with combination regimens proposed for some of them mainly based on expert opinion [17]. 

In addition to the absence of clear evidence of clinical benefit for the treatment of IMI, there are reports suggesting potentially antagonistic interactions when combining azoles and amphotericin B [18,19,20]. Furthermore, combination therapies enhance the risk of drug–drug interactions and toxicities and increase costs.

Considering the lack of conclusive data on the utility of combination antifungal treatment of IA and non-IA IMI despite intensive clinical research conducted during the last two decades, we sought to describe our single-center, real-life experience on the indications and outcomes of combination antifungal therapy in a ten-year cohort of allogeneic HCT recipients.

## 2. Materials and Methods

### 2.1. Study Design

This study is part of a retrospective single-center cohort study, which was conducted among all consecutive adult (age ≥ 18-year-old) allogeneic HCT recipients who had an HCT performed between 1 January 2010 and 1 January 2020 at the University Hospital of Geneva in Switzerland, as previously reported [5]. For this study, only patients who had a post-HCT diagnosis of proven or probable IMI, according to the latest international diagnostic criteria, and with at least 1 year of follow-up (unless patients died before that) after their IMI diagnosis were included [21]. This study was approved by the local ethics committee (2020-01072).

### 2.2. Study Objectives

The primary objective of this study was to describe the frequency and indications of combination antifungal treatment for the management of IMI after an allogeneic HCT. As secondary objectives, we assessed the duration of combination antifungal treatment and 12-week and 1-year overall survival, stratifying patients according to the type of IMI (IA vs. non-IA). 

### 2.3. Patient Identification and Data Collection

Allogeneic HCT recipients were identified via the institutional HCT database. The following variables were retrieved directly through the HCT database: (i) demographics: age, gender, (ii) underlying hematologic diagnosis leading to HCT, (iii) HCT-related variables: conditioning regimen (myeloablative, reduced intensity), HCT donor type (matched related, matched/mismatched unrelated, or haplo-identical donor), HCT source (bone marrow, peripheral blood stem cells), and cytomegalovirus (CMV) serology status of donor (D) and recipient (R), (iv) post-HCT complications: ≥grade 2 acute and chronic GvHD. A detailed chart review was performed for all patients to identify those patients with proven and probable IMI, as previously described [5]. To confirm that all IMI cases were captured, we compared this list to the dataset provided by the mycology laboratory of our institution. All identified cases were carefully reviewed by two investigators (R.R. and D.N.) to confirm a proven or probable IMI diagnosis. For all patients with a proven and probable IMI, the following additional data were captured: IMI infection sites, pathogens, administered antifungal therapy (agents and duration), and 12-week and 1-year all-cause mortality.

### 2.4. Definitions

Proven and probable IMIs were defined based on the latest international consensus guidelines [21]. In brief, proven IMI was defined as the presence of histopathologic or other microscopic changes in specimens obtained by biopsy or needle aspiration in which hyphae were observed accompanied by tissue damage. Probable IMI was considered in the presence of a predisposing host factor (in this case, the post-HCT context) accompanied by clinical/radiological and mycological evidence of IMI. For proven cases, tissue or other normally sterile material obtained by biopsy/needle aspiration was sent for fungal culture and *Aspergillus fumigatus* as well as panfungal PCR testing. For probable cases, respiratory secretions (sputum, bronchoalveolar lavage (BAL), aspirate) or sinus aspirates were sent for fungal culture and *Aspergillus fumigatus* as well as panfungal PCR testing. Elisa galactomannan testing in serum and/or BAL was performed in all cases in which IMI was suspected and was considered positive at a cutoff of an optical density index of 0.5 and when fulfilling the mycological criterion for IA when ≥1.0 in serum or BAL or ≥0.7 in serum and ≥0.8 in BAL, or ≥1.0 in cerebrospinal fluid. Patients having only galactomannan positivity as their mycological evidence of IMI were classified as IA due to *Aspergillus* spp. (species identification not available). 

Combination therapy was defined as the concomitant administration of two or more antifungal agents from different classes for a minimum of 3 consecutive days. Considering the multiple changes in antifungal therapy observed in this cohort, and in order to facilitate analyses, the following considerations were made: (i) Patients who received only monotherapy were considered as a separate “monotherapy group”. (ii) Patients who received combination antifungal therapy were divided into two groups: (a) “<50% combination therapy group”, including patients whose treatment course included combination therapy in <50% of the overall treatment duration, and (b) “≥50% combination therapy group”, including patients whose treatment course consisted of ≥50% of combination treatment. 

### 2.5. Statistical Analysis

Categorical variables were summarized as numbers and percentages and were compared using Fisher’s exact test. Continuous variables were presented as medians and interquartile ranges and were compared using a two-sided t-test or Mann–Whitney depending on their distribution. Kaplan–Meier survival curves were used to depict survival probabilities from the diagnosis of an IMI up to 12 weeks and 1 year post-diagnosis. The log-rank test was used to compare survival probabilities among strata. A *p*-value of 0.05 or less was considered statistically significant. All analyses were performed using R Statistical Software (version 4.0.3 (2020); R Foundation for Statistical Computing, Vienna, Austria).

## 3. Results

### 3.1. Study Population and Baseline Patient Characteristics with IMI

From 1 January 2010 through 1 January 2020, 515 patients received an allogeneic HCT at our institution. Of them, 47 patients (9.1% of all HCT patients) were diagnosed with 48 proven or probable IMIs after transplantation (46 patients had 1 IMI diagnosis, while 1 patient had 2 separate IMI diagnoses), with 33/48 (68.7%) cases of IA and 15/48 (31.3%) cases of non-IA IMI. Table 1 summarizes the baseline patient characteristics at the time of IMI diagnosis. The predominant underlying hematologic condition leading to HCT was acute leukemia (59% of cases). Fifty-five percent (55%) of patients had a matched unrelated donor. Moderate to severe GvHD was present in 66% of the patients. There were no significant differences in baseline attributes between IA and non-IA IMI cases, although there was a trend towards an increased frequency of leukemia as a baseline condition in non-IA IMI patients compared to IA (82% vs. 49%, *p* = 0.06)

### 3.2. Timing, Pathogens, and Clinical Characteristics of IMI

Details about IMI features are presented in Table 2. IMI was diagnosed at a median of 189 days after transplantation: 173 days for IA and 218 days for non-IA IMI (*p* = 0.84). Of the 48 IMI cases, there were 37 (77%) probable and 11 (23%) proven IMIs. There was a trend for a higher proportion of proven diagnoses among non-IA IMI vs. IA IMI cases (15% vs. 40%, *p* = 0.07). There were 11 (23%) disseminated IMIs (>1 site involved): 6/33 (19%) and 5/15 (34%) cases of IA and non-IA IMI, respectively (*p* = 0.28). The primary infected site was the lungs in 91% of cases, followed by the sinuses in 11%. Skin and soft tissue or brain involvement was present in 11 and 4% of cases, respectively. 

Precise identification of the involved pathogen was possible for 17/33 (55%) IA cases vs. 100% in the non-IA IMI group (*p* < 0.01). Among patients with IA, microbiological analyses revealed the presence of *Aspergillus fumigatus* (13), *A. terreus* (3), *A. ustus* (3), and *A. niger* (2), with four patients having more than one *Aspergillus* species recovered from clinical specimens (two cases of mixed *A. fumigatus* and *A. terreus*, and two cases of *A. terreus* and *A. niger*). In the non-IA IMI group, analyses revealed the following pathogens: *Rhizomucor* spp. (7), *Rhizopus* spp. (2), *Lichtemia* spp. (1), *Fusarium* spp. (2), *Scedosporium* spp. (1), *Scopulariopsis* spp. (1), *Schizophyllum* spp. (1), *Alternaria* spp. (1). Mixed non-IA IMI included one case of *Rhizomucor* spp. and *Rhizopus* spp. infection and one case of *Rhizomucor* spp. and *Scopulariopsis* spp. infection. There were no patients presenting concomitant IA and non-IA IMI, although there was a single patient in whom a mixed infection with *A. fumigatus* and *A. terreus* was diagnosed, followed by a *Rhizomucor* spp. IMI 18 days later. 

### 3.3. Antifungal Treatment of IMI

All forty-seven (47) patients received antifungal therapy for their IMI. A total of 23 (49%) patients received pure monotherapy treatment, while 24 (51%) received at least one course of antifungal combination therapy: 4/47 (9%) with pure combination treatment, and 20/47 (42%) with mixed monotherapy/combination treatment, as detailed in Table 3.

#### 3.3.1. Antifungal Combination Therapies Administered

Combination treatment was started at a median of 8 (IQR: 2, 19) days post-IMI diagnosis. Almost one third of patients received a minimum of a week-long combination treatment during the first 4 weeks of their treatment course, more frequently in patients with non-IA IMI (7, 47%) compared to IA IMI (7, 21%; *p* = 0.09). Among patients that received combination treatment, there was a median of two combination courses administered per patient (range 1–4). The clinical reasons motivating the selection of an initial combination antifungal therapy included severe IMI (18, 38%), lack of antifungal susceptibility data (14, 30%), lack of pathogen identification (5, 11%), and combination treatment until reaching a therapeutic azole serum level (6, 13%). The most common combination treatments were azole/liposomal amphotericin-B (28%) and liposomal amphotericin-B/echinocandin (21%). Combination treatment was administered cumulatively for a median duration of 28 days (IQR: 7, 47): 14 (IQR: 6, 50) days for IA and 28 (IQR: 21, 34) days for non-IA IMI (*p* = 0.18). Although the differences were non-statistically significant, all types of antifungal combinations were generally longer in duration in non-IA IMI, with the exception of the liposomal amphotericin B/echinocandin combination regimen that showed a trend of an increased duration in the IA vs. non-IA IMI group (33 vs. 6 days, *p* = 0.08). 

#### 3.3.2. Antifungal Combination Therapy Changes

There were 163 treatment courses administered: 115 in the IA group and 48 in the non-IA IMI group. Overall, there were, on average, three treatment changes per patient (range 0–8). The variability and complexity of antifungal treatment of IMI are presented in Figure 1, which shows the number of patients receiving combination therapy and monotherapy from the beginning of their antifungal therapy through week 12 of treatment, with all treatment changes performed between those time points per IMI treated. 

The reasons prompting the selection between monotherapy and combination therapy for the 163 treatment courses are described in detail in Table 4. Clinical efficacy reasons were the principal motivation for 42/48 (88%) combination treatment prescriptions, compared to 80/115 (70%) monotherapy prescriptions (*p* = 0.02). More precisely, 19 (40%) of the combination courses were selected to target specific fungal pathogens (compared to 21% of monotherapy courses, *p* < 0.01). This concerned specifically 19 prescriptions in 12 patients, of which 5 had IA and 7 had non-IA IMI. Focusing on the non-IA IMI cases, two were disseminated *Mucorales* spp. IMI, while three involved mixed and difficult to treat pathogens: one *Scedosporium* spp. infection, one mixed *Scopulariopsis* spp. and *Mucorales* infection, and one *Mucorales* infection after *A. fumigatus* and *A. terreus* IA. The selection of combination treatment as targeted therapy in five cases of IA revealed that three concerned disseminated *A. ustus* infection and two mixed *A. fumigatus* and *A. terreus* IA. In 14 (30%) courses, combination treatment was used to treat stable or progressive IMI (vs. 11% of monotherapy courses, *p* < 0.01) and in 9 (19%) courses to address low azole serum drug levels (vs. 3% of monotherapy courses, *p* < 0.01). Drug toxicity was the motivation for 10% of combination treatment prescriptions, compared to 30% in monotherapy prescriptions (*p* < 0.01). Finally, there were no logistical issues taken into consideration while prescribing combination therapies, while this was the case in 4% of monotherapy prescriptions. 

### 3.4. Combination Antifungal Treatment and IMI Mortality

All-cause mortality was 23% at 6 weeks, 30% at 12 weeks, 51% at 24 weeks, and 64% at 1 year post-IMI diagnosis. All-cause 12-week (Figure 2; logrank = 0.67) and 1-year post-IMI diagnosis survival probabilities (Appendix A; logrank = 0.92) for IA and non-IA IMI patients did not significantly differ. The median time from IMI diagnosis to death was 130 days: 123 days in the IA group vs. 150 days in the non-IA IMI group (logrank = 0.94). Patients receiving ≥50% of their treatment as combination treatment had worse 12-week survival compared to monotherapy and patients receiving <50% of their treatment as combination (logrank = 0.05, Figure 3a). Notably, while there was no significant difference in 12-week survival based on the mode of treatment in the IA group (logrank = 0.63, Figure 3b), non-IA IMI patients receiving ≥50% of their treatment as combination had worse 12-week survival compared to monotherapy and patients receiving <50% of their treatment as combination (logrank = 0.03, Figure 3c).

## 4. Discussion

This single-center, retrospective cohort study provides insights on contemporary treatment modalities of IMI after allogeneic HCT. In this one of few real-life studies examining the motivations leading to combination antifungal therapy initiation in this population, we report that combination therapy prescription is relatively common, with 50% of allogeneic HCT recipients with IMI receiving combination treatment [22,23]. Notably, almost one third of patients received at least 7 days of antifungal combination therapy during the first month of treatment for a median of two combination courses per IMI. Despite the lack of relevant recommendations, these high rates of combination antifungal treatment administration underscore the severity of clinical presentations of IMI in high-risk allogeneic HCT recipients and the degree of clinical concern leading to prescriptions of treatments not always strictly following current consensus guidelines [16,17,24,25]. The latter is further supported by a trend showing that patients treated for a non-IA IMI, an infection historically associated with high mortality rates, were more likely to receive a minimum of a week-long combination treatment during the first 4 weeks of their treatment course.

In addition to its frequent administration, we observed a large variability in the types, timing, duration, and reasons for initiation of and changes in combination antifungal treatment, as depicted in Figure 1. These findings underscore the complexity of the treatment of IMI in high-risk hematology patients, but also the lack of conclusive relevant guidance. Through the years, conflicting retrospective and in vitro data have created confusion when it comes to the real utility of antifungal combination treatment [8,9,10,11,12,13,14,16,17]. Similarly, the only placebo-controlled clinical trial to compare voriconazole with anidulafungin for the first 2 weeks of treatment for IA vs. voriconazole alone failed to definitively answer the question of whether more is better [9]. Hence, consensus guidelines do not endorse combination treatment for the treatment of IA, although it is suggested that a potential benefit in specific cases may be obtained by combining more than one antifungal class in the management of non-IA IMI [16,17,24,25]. The relative rarity of those infections, along with the large variability of pathogens and susceptibility profiles, has hindered, and will likely continue to affect, the conduction of reasonably sized and realistic clinical trials to answer this very question for different types of IMI. Furthermore, parameters such as the timing and duration of combination antifungal treatment and/or the types of antifungal classes to be combined further add to the complexity of the management of those infections. Hence, and based on the existing conflicting retrospective data and lack of definitive guidelines, clinicians remain confused as to whether combining more than one agent in critically ill patients with difficult to treat infections is better than monotherapy. In such situations, and out of desperation, they may prescribe variable combinations for variable durations of time, based on their clinical experience, experts’ opinion, in vitro susceptibility testing results, and synergy data. The above are reflected in our observations, showing inconsistencies and a large variability in the co-administration of different antifungal agents, and the timing and duration of treatment. This gap between a real clinical need for better guidance and lack of data availability urgently calls for more data and uniform guidelines in an effort to provide homogeneous and standardized recommendations for the treatment of difficult to treat IMI.

Clinical efficacy issues dominated, indeed, the initial choice of combination therapy, leading to the use of a striking 88% of combination treatment prescriptions. Stable or progressive IMIs based on clinical and radiological criteria have been classified as treatment failure according to international consensus definitions, and this was the motivation of a substantial number of combination therapy prescriptions (30% of all courses) [26]. This comes as no surprise given that disseminated and/or non-IA IMI, traditionally considered difficult to treat, accounted for 44% of cases in this series. While solid evidence is lacking, combination therapies have been reported to improve outcomes in some cases of non-IA IMI [17]. Hence, clinicians caring for high-risk patients with severe IMI appear to be more prone to initiate combination antifungal treatment, in an effort to optimize clinical outcomes. Combination therapy was also preferred as the targeted treatment of IMI in about 40% of prescriptions, including five cases of IA. These cases included three cases of disseminated *A. ustus* infection and two mixed *A. fumigatus* and *A. terreus* IA. *A. terreus* IA has been linked to a dismal prognosis, while *A. ustus* with disseminated breakthrough and difficult to control IA with devastating outcomes probably explaining why combination therapy was preferred in these cases [27,28].

Combination therapy was administered to avoid azole sub-therapeutic concentrations in up to 19% of prescriptions. Considering the time to steady state for most broad-spectrum azoles, absorption concerns in high-risk patients with mucositis or gastrointestinal GvHD, and multiple drug interactions, at times affecting drug absorption and metabolism, clinicians may be concerned about adequate azole dosing and plasma concentrations. The latter are even more pertinent when treating severe life-threatening infections, such as IMI, the course of which can be fulminant in the setting of severe immunosuppression, and treatment delays have been associated with worse outcomes [29]. Thus, our data suggest that in addition to other reasons, in clinical practice, combination treatment may be administered as a bridge to effective targeted treatment initiation, at least until therapeutic azole concentrations are assured to optimize patient care.

The above are better reflected by the significantly higher mortality observed in patients who received ≥50% of their treatment as different combinations of antifungals. This was mainly due to patients with non-IA IMI receiving ≥50% of treatment as combination. We hypothesize that clinicians are more likely to prescribe longer treatment courses of combination antifungal treatment to sicker patients with worse overall prognosis, who are more likely to die. Hence, the higher mortality observed in this patient group does not necessarily imply treatment failure, but rather patient selection biases, with patients with severe or refractory disease receiving more intensive antifungal therapies, without, however, the desired clinical outcomes. Notably, mortality was almost superimposable between patients who were treated with monotherapy and those whose treatment included <50% of combination treatment.

In addition to the lack of benefit on clinical outcomes by combination therapy, potential drug interactions and associated toxicities may occur when combining more agents from different classes, increasing the overall toxicity rates and healthcare-related costs. In our study, toxicity was a concern taken into account in 24% of prescriptions in daily practice and favored the initiation of monotherapy in the majority of such cases. While our study does not demonstrate significant imbalances in terms of drug interactions or logistic issues such as costs when selecting between monotherapy and combination therapy, the numbers of cases represented in these categories are too small to draw any further conclusions. Liposomal amphotericin-B-containing regimens were used in about 60% of all cases (in combination with either an echinocandin, an azole, or both). Despite concerns for renal toxicity, liposomal amphotericin-B provides a broader coverage spectrum compared to most azoles including *Mucorales* species and difficult to treat filamentous fungi other than *Mucorales*, which have been shown to be implicated in a substantial number of IMIs after HCT and accounted for 31% of all IMIs in our study [3,6]. Although lower doses of liposomal amphotericin-B have been associated with improved survival, likely due to lower rates of nephrotoxicity, the dose, duration, and different combinations of liposomal amphotericin-B for the treatment of non-IA IMI remain to be defined [30].

The main limitations of this study are its retrospective design and the small number of IMIs included based on a single-center cohort. In addition, and while this study cannot definitively answer the question of the real benefit of combined antifungal therapy, we were not able to make any further conclusions on the utilization of resources, associated costs, and potential additive toxicities, as a result of combining more than one agent together. However, our data demonstrate the complexities and inconsistencies observed in the administration of combination antifungal therapy in clinical practice and the real ongoing need for better treatment options for difficult to treat infections in high-risk fragile patient populations, such as allogeneic HCT recipients.

## Figures and Tables

**Figure 1 jof-07-00811-f001:**
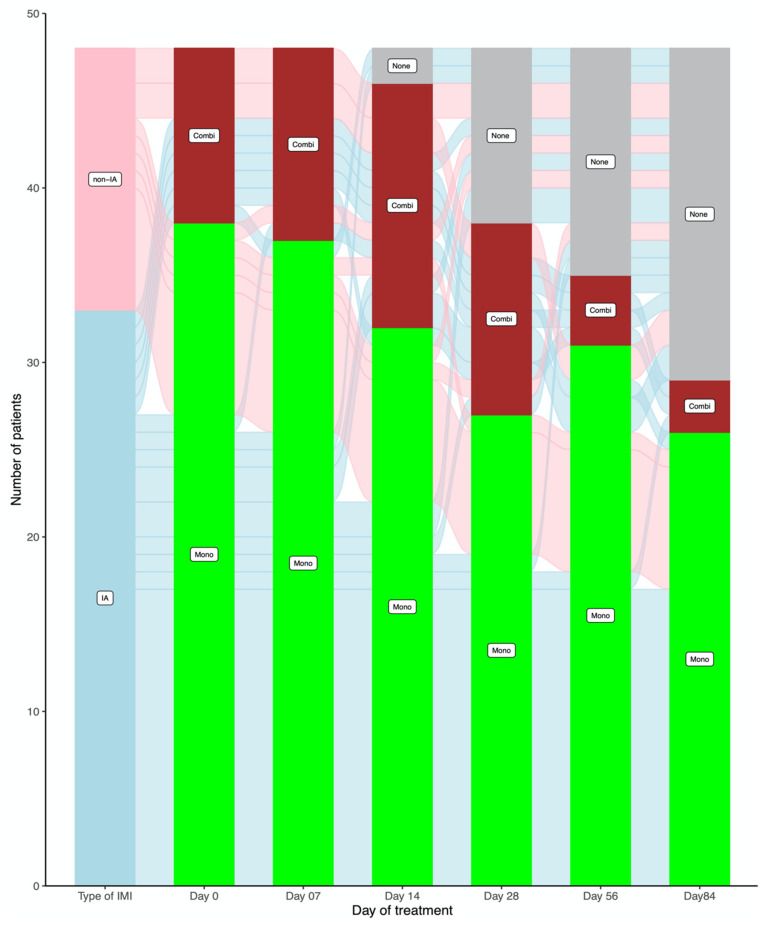
Alluvial plot summarizing the administration of antifungal therapy and treatment changes between monotherapy and combination antifungal treatment from baseline and during the first 12 weeks after the diagnosis of an invasive mold infection. IMI: invasive mold infection, IA: invasive aspergillosis, non-IA: non-IA IMI, Mono: monotherapy, Combi: combination therapy, None: no treatment.

**Figure 2 jof-07-00811-f002:**
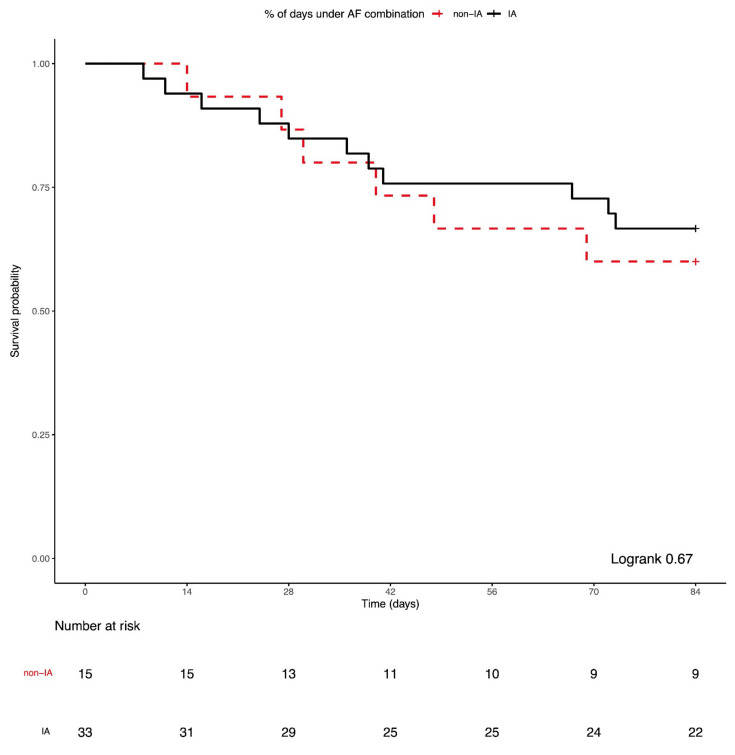
Kaplan–Meier survival curves of all-cause 12-week mortality after the diagnosis of an invasive mold infection (IMI) according to IMI type: invasive aspergillosis (IA) vs. non-IA IMI.

**Figure 3 jof-07-00811-f003:**
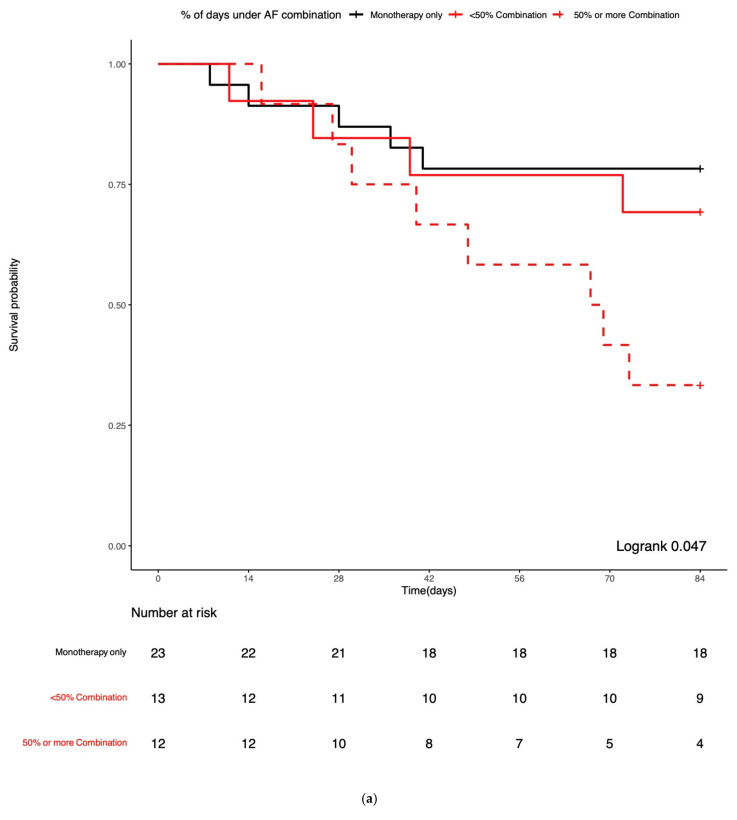
Kaplan–Meier survival curves of all-cause 12-week mortality after the diagnosis of an invasive mold infection (IMI) according to the number of days of combination therapy administration: Monotherapy only, and <50% and ≥50% of combination treatment of their entire treatment course for: (**a**) all patients with IMI, (**b**) patients with invasive aspergillosis (IA), and (**c**) patients with non-IA IMI. IMI: invasive mold infection, IA: invasive aspergillosis, non-IA: non-IA IMI, AF: antifungal treatment.

**Table 1 jof-07-00811-t001:** Baseline characteristics of 47 allogeneic hematopoietic transplant recipients diagnosed with 48 invasive mold infections.

	All Patients *N* = 47	IA *N* = 33 ^1^	Non-IA IMI *N* = 15 ^1^	*p*-Value
Demographics				
Age at IMI diagnosis, median (IQR)	57 (41–61)	57 (42–61)	56 (43–61)	0.72
Sex, Female	19 (40)	12 (36)	7 (47)	0.53
Underlying disease leading to HCT				
Acute leukemia ^1^	27 (59)	16 (49)	12 (82)	0.06
Myelodysplastic syndrome	6 (12)	5 (15)	1 (6)	0.65
Lymphoma	6 (12)	5 (15)	1 (6)	0.65
Other ^2^	8 (17)	7 (21)	1 (6)	0.41
HCT-related characteristics				
Conditioning regimen, myeloablative	9 (19)	4 (12)	5 (33)	0.12
HCT source, bone marrow	8 (17)	5 (15)	3 (20)	0.69
Donor-related characteristics				
Matched related ^1^	10 (22)	8 (25)	3 (20)	1
Matched unrelated	26 (55)	16 (48)	10 (67)	0.35
Mismatched related	3 (6)	3 (9)		1
Haplo-identical	8 (17)	6 (18)	2 (13)	0.54
CMV D/R serologic status				
D-R-	10 (22)	7 (21)	3 (20)	1
D-R+/D+R+ ^1^	31 (66)	22 (67)	10 (67)	1
D+R-	6 (12)	4 (12)	2 (13)	1
GvHD				
Acute GvHD ≥ grade 2 ^1^	31 (66)	23 (70)	9 (60)	0.53
Chronic GvHD	12 (24)	9 (27)	3 (20)	0.73

Continuous variables are presented in this table as medians and interquartile ranges and categorical variables as number of observations and percentages. IMI: invasive mold infection, IA: invasive aspergillosis, IQR: interquartile range, HCT: hematopoietic cell transplantation, CMV: Cytomegalovirus, D: donor, R: recipient, GvHD: graft-versus-host disease. ^1^ One patient was diagnosed with both IA and non-IA infection post-transplantation and was counted in both IA and non-IA IMI groups. ^2^ Other hematologic diseases included in the IA group: aplastic anemia (1), chronic myeloid leukemia (1), myeloproliferative syndrome (4), multiple myeloma (1); in the non-IA IMI group: myeloproliferative syndrome (1).

**Table 2 jof-07-00811-t002:** Characteristics and outcomes of 48 invasive mold infections in 47 allogeneic hematopoietic cell transplant recipients.

	All Patients *N* = 47	IA *N* = 33 ^1^	Non-IA IMI *N* = 15 ^1^	*p*-Value
Timing of IMI post-HCT				
Post-HCT day IMI diagnosis, median (IQR)	189 (19–376)	173 (21–364)	218 (16–344)	0.84
EORTC–MSGERC Classification ^2^				
Probable IMI	37 (77)	28 (85)	9 (60)	0.07
Proven IMI	11 (23)	5 (15)	6 (40)	0.07
Extent of infection				
Single site	36 (77)	27 (82)	10 (66)	0.28
Disseminated (>1 sites)	11 (23)	6 (19)	5 (34)	0.28
Sites of infection ^3^				
Lung	43 (91)	30 (91)	13 (87)	0.64
Sinus	5 (11)	3 (9)	2 (13)	0.64
Brain	2 (4)	1 (3)	1 (6)	0.53
Skin/soft tissues	5 (11)	2 (6)	3 (20)	0.32
Other ^4^	6 (12)	4 (12)	2 (13)	1
IMI pathogens				
*Aspergillus* spp ^5^	33 (69)	33 (100)		
*Mucorales* species	9 (19)		9 (60)	
*Fusarium* spp.	2 (4)		2 (13)	
Other ^6^	4 (8)		4 (27)	
Outcomes ^7^				
Death by week 6	11 (23)	7 (21)	4 (29)	0.70
Death by week 12	14 (30)	8 (24)	6 (40)	0.30
Death by week 24	24 (51)	17 (52)	7 (50)	1
Death by day 365	30 (64)	20 (61)	10 (71)	0.53
Days from IMI diagnosis to death, median (IQR)	130 (45–494)	123 (67-497)	150 (44–390)	0.94

Continuous variables are presented in this table as medians and interquartile ranges and categorical variables as number of observations and percentages. IMI: invasive mold infection, IA: invasive aspergillosis, IQR: interquartile range, HCT: hematopoietic cell transplantation, EORTC/MSGERC: European Organization for Research and Treatment of Cancer and the Mycoses Study Group Education and Research Consortium. ^1^ One patient was diagnosed with both IA and non-IA infection post-transplantation and was counted in both IA and non-IA IMI groups. ^2^ Latest (2020) updated consensus definitions by EORTC/MSGERC (18). ^3^ More than one site could be involved per patient. ^4^ Other included a positive blood culture (1), abdominal (1), or other/non-specified (2) infection in the IA group, and other/non-specified (2) infections in the non-IA IMI group. ^5^ Diagnosis of IA was based only on galactomannan detection in serum and/or bronchoalveolar lavage in 16/33 (48%) IA cases. ^6^ Other pathogens included *Hormographiella aspergillata* (1), *Scedosporium apiospermum* (1), *Schizophyllum commune* (1), and *Alternaria* spp. (1). ^7^ One patient diagnosed with both IA and non-IA IMI after transplantation was counted only in the non-IA IMI outcome group (non-IA IMI diagnosed after IA).

**Table 3 jof-07-00811-t003:** Management of IMI with emphasis on combination treatments.

	All Patients *N* = 47	IA *N* = 33 ^1^	Non-IA IMI *N* = 15 ^1^	*p*-Value
Time from IMI to treatment initiation, median (IQR)	0 (1–3)	0 (0–2)	2 (0.5–4)	0.08
Type of treatment				
Monotherapy only	23 (49)	16 (48)	7 (47)	1
Combination therapy only	4 (9)	2 (7)	2 (13)	0.57
Monotherapy/combination treatment	20 (42)	15 (45)	6 (40)	0.75
Treatment duration				
Overall treatment duration in days, median (IQR)	112 (40–208)	112 (60–195)	99 (28–250)	0.30
Time to combination treatment initiation				
Post-IMI diagnosis day, median (IQR)	8 (2–19)	9 (2–34)	6 (2 -8)	0.46
Administration within first 7 days from IMI diagnosis	11 (23)	7 (21)	4 (27)	0.70
Administration within first 14 days from IMI diagnosis	17 (36)	10 (30)	7 (47)	0.33
Administration for ≥7 days during first 28 days from IMI diagnosis	14 (30)	7 (21)	7 (47)	0.09
Courses of combination treatment				
Combination courses per patient, median (IQR)	2 (1–3)	1 (1–3)	2 (1–2)	0.87
Patients with 1 combination course	12 (26)	9 (27)	3 (20)	0.67
Patients with >1 combination courses	12 (26)	8 (24)	5 (33)	0.67
Indication for combination treatment initiation ^2^				
Severe IMI	18 (38)	10 (30)	9 (60)	0.54
Pathogen identification lacking	5 (11)	1 (3)	4 (27)	0.14
Antifungal susceptibility lacking	14 (30)	10 (30)	4 (27)	0.33
Combination until therapeutic azole serum concentration	6 (13)	5 (15)	1 (7)	0.38
Types of treatment combinations ^2^				
Azole/echinocandin	8 (17)	5 (15)	3 (20)	1
Azole/amphotericin-B	13 (28)	8 (24)	6 (40)	0.49
Amphotericin B/echinocandin	10 (21)	8 (24)	2 (13)	0.43
Azole/amphotericin-B/echinocandin	5 (11)	3 (9)	2 (13)	1
Duration of combination treatment				
Cumulative days, median (IQR)	28 (7–47)	14 (6–50)	28 (21–34)	0.18
Azole/echinocandin days, median (IQR)	11 (8–15)	12 (8–16)	10 (9–13)	0.78
Azole/amphotericin-B days, median (IQR)	12 (5–32)	7 (5–13)	25 (13–37)	0.19
Echinocandin/amphotericin-B days, median (IQR)	19 (7–65)	33 (11–74)	6 (5–7)	0.08
Azole/echinocandin/amphotericin-B days, median (IQR)	10 (8–17)	10 (9–15)	12 (9–14)	0.80

Continuous variables are presented in this table as medians and interquartile ranges and categorical variables as number of observations and percentages. IMI: invasive mold infection, IA: invasive aspergillosis, IQR: interquartile range, HCT: hematopoietic cell transplantation. ^1^ One patient was diagnosed with both IA and non-IA IMI post-transplantation and was counted in both IA and non-IA IMI groups. ^2^ More than one possible per patient.

**Table 4 jof-07-00811-t004:** Reasons for monotherapy and combination course initiation.

	All Treatment Courses *N* = 163	Monotherapy Courses *N* = 115	Combination Courses *N* = 48	*p*-Value
Treatment initiation reasons ^1^				
Clinical efficacy	122 (75)	80 (70)	42 (88)	0.02
IA suspicion	46 (28)	36 (31)	10 (21)	0.19
Non-IA IMI suspicion	12 (7)	8 (7)	4 (8)	0.74
Switch to targeted therapy	43 (26)	24 (21)	19 (40)	<0.01
Low azole concentration in serum	10 (6)	3 (3)	9 (19)	<0.01
Stable or progressive IMI	27 (17)	13 (11)	14 (30)	<0.01
Improvement in IMI	4 (2)	4 (3)		
Toxicity	39 (24)	34 (30)	5 (10)	<0.01
Renal toxicity	9 (6)	8 (7)	1 (2)	0.28
Liver toxicity	17 (10)	14 (12)	3 (6)	0.40
Neurologic toxicity	4 (2)	4 (3)		
Cutaneous toxicity	2 (1)	2 (3)		
QT interval prolongation	1 (1)	1 (1)		
Drug interactions	6 (4)	5 (4)	1 (2)	0.67
Increased azole concentration in serum	1 (1)	1 (1)		
Logistical reasons	6 (4)	6 (5)		
Change from IV to PO	4 (2)	4 (3)		
Costs and insurance coverage	2 (1)	2 (3)		

Continuous variables are presented in this table as medians and interquartile ranges and categorical variables as number of observations and percentages. IMI: invasive mold infection, IA: invasive aspergillosis, IV: intravenous, PO: oral. ^1^ More than one possible per patient.

## Data Availability

Dataset available upon request to the corresponding author.

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
