# Peer review of "Real-Life Considerations on Antifungal Treatment Combinations for the Management of Invasive Mold Infections after Allogeneic Cell Transplantation"

_jof, 2021, doi:10.3390/jof7100811_

Round 1

Reviewer 1 Report

In this single-center retrospective cohort study the authors have  analyzed indications, timing, and outcomes of combination antifungal therapy in post- hematopoietic cell transplantation (HCT ) mold infections. Overall, the study is very comprehensively done. However, I have some comments:

  1. Introduction is not informative enough.
  2. It is necessary to describe in this article the criteria used to differentiate proven vs probable vs probable IMI, just merely citing an article is not enough.
  3. It is also not clear how microbiological analyses were done-a detail description in the method section is needed.
  4. Line# 22: “Among 47 patients, 48 IMI episodes were identified” what does it mean
  5. Line # 122: “47 patients were diagnosed with 48 proven or probable IMI “ what does that mean?
  6. In table 1, what are values in the parenthesis? e.g. Lymphoma 6 (12). Is it the % of patients? If so, then it's worth mentioning.

Author Response

We would like to thank the reviewer for their insightful comments and suggestions. A point-by-point response to all reviewer’s comments can be found below.

  1. Introduction is not informative enough

Response:

We thank the reviewer for this point. We have now added in our Introduction a short paragraph describing the potential dangers of combination treatments as this was not the case with the originally submitted manuscript. In our opinion this will help the readers of the journal to have a more global and balanced introduction to the utility of antifungal in combination (lines 64-69). A paragraph regarding these issues has been also added in the discussion (lines 392 to 400).

  1. It is necessary to describe in this article the criteria used to differentiate proven vs probable vs probable IMI, just merely citing an article is not enough.

Response:

We thank the reviewer for the comment. A brief description of the latest consensus criteria for the diagnosis of invasive mold diseases was included in lines 100 to 104.

  1. It is also not clear how microbiological analyses were done-a detail description in the method section is needed.

Response:

This is a very pertinent comment. We have included an extensive description of the microbiological methods used in our institution in lines 105 to 116.

  1. Line# 22: “Among 47 patients, 48 IMI episodes were identified” what does it mean

Among the 47 included patients 46 had 1 IMI episode while one patient had 2 IMI episodes giving in total 48 IMI infections.

Response:

This is now clarified in the abstract section.

  1. Line # 122: “47 patients were diagnosed with 48 proven or probable IMI “ what does that mean?

Response:

This is now clarified in the results section: Among the 47 included patients 46 had 1 IMI episode while one patient had 2 IMI episodes giving in total 48 IMI infections.

  1. In table 1, what are values in the parenthesis? e.g. Lymphoma 6 (12). Is it the % of patients? If so, then it's worth mentioning.

Response:

We agree with the reviewer that the meaning of the numbers in parentheses in tables was not clear. We have now added a footnote in each table, which clarifies the meaning of all numbers in and outside parentheses in tables.

Reviewer 2 Report

The authors present a retrospective study outlining the use of antifungal therapy in the management of invasive mold infections in patients who have had HCT. The study revealed that 51% of patients received combined antifungal therapy. The authors explored what caused clinicians to use combination antifungal therapy. The patients who received combination therapy include those with severe disease, the lack of antifungal susceptibility data or identification of fungi. The authors also examined mortality differences at different time-point in the different invasive mycoses studied.

The authors highlight how their findings underscore the severity of disease usually encountered in post HCT patients and the clinical concern resulting in guidelines not being followed. The study discusses and highlights the difficulty in getting a reasonably-sized large number to address clinical studies in this space. 

The study highlights the need to address these challenges and the need for better treatment options,

The limitation of the study are highlighted

Minor points

1. Table 4

Under switch to targetted therapy there are more all treatment course 45 that 43 (the total of combination therapy and monotherapy) could the authors please explain.

2. The author mentioned there was one patient that had IA and later non-IA does this explain the discrepancy seen on table 1, under acute leukemia, matched related donor characteristics and  D-R+/D+R+ where the sum of the number of IA and non-IA patients exceed the All patients by 1? if so a footnote explaining that would be helpful.

Otherwise, a well-written manuscript highlighting the challenges in real life.

Author Response

We would like to thank the reviewer for their insightful comments and suggestions. A point-by-point response to all reviewer’s comments can be found below.

  1. Table 4. Under switch to targeted therapy there are more all treatment course 45 that 43 (the total of combination therapy and monotherapy) could the authors please explain.

Response:

We thank the reviewer for this remark. Indeed this was a typo that is now corrected in Table 4.

  1. The author mentioned there was one patient that had IA and later non-IA does this explain the discrepancy seen on table 1, under acute leukemia, matched related donor characteristics and  D-R+/D+R+ where the sum of the number of IA and non-IA patients exceed the All patients by 1? if so a footnote explaining that would be helpful.

Response:

This discrepancy is due to one patient that was diagnosed with IA and non-IA IMI and it’s attributes are counted in both IA and non-IA groups in the table. We have now included one footnote to clarify this point.